# Vagal Threshold Determination during Incremental Stepwise Exercise in Normoxia and Normobaric Hypoxia

**DOI:** 10.3390/ijerph17207579

**Published:** 2020-10-19

**Authors:** Filip Neuls, Jakub Krejci, Ales Jakubec, Michal Botek, Michal Valenta

**Affiliations:** Department of Natural Sciences in Kinanthropology, Faculty of Physical Culture, Palacky University Olomouc, 771 11 Olomouc, Czech Republic; jakub.krejci@upol.cz (J.K.); ales.jakubec@upol.cz (A.J.); michal.botek@upol.cz (M.B.); michal.valenta@upol.cz (M.V.)

**Keywords:** autonomic nervous system, vagal withdrawal, simulated altitude, exercise intensity, saturation

## Abstract

This study focuses on the determination of the vagal threshold (T_va_) during exercise with increasing intensity in normoxia and normobaric hypoxia. The experimental protocol was performed by 28 healthy men aged 20 to 30 years. It included three stages of exercise on a bicycle ergometer with a fraction of inspired oxygen (FiO_2_) 20.9% (normoxia), 17.3% (simulated altitude ~1500 m), and 15.3% (~2500 m) at intensity associated with 20% to 70% of the maximal heart rate reserve (MHRR) set in normoxia. T_va_ level in normoxia was determined at exercise intensity corresponding with (M ± SD) 45.0 ± 5.6% of MHRR. Power output at T_va_ (PO_th_), representing threshold exercise intensity, decreased with increasing degree of hypoxia (normoxia: 114 ± 29 W; FiO_2_ = 17.3%: 110 ± 27 W; FiO_2_ = 15.3%: 96 ± 32 W). Significant changes in PO_th_ were observed with FiO_2_ = 15.3% compared to normoxia (*p* = 0.007) and FiO_2_ = 17.3% (*p* = 0.001). Consequentially, normoxic %MHRR adjusted for hypoxia with FiO_2_ = 15.3% was reduced to 39.9 ± 5.5%. Considering the convenient altitude for exercise in hypoxia, PO_th_ did not differ excessively between normoxic conditions and the simulated altitude of ~1500 m, while more substantial decline of PO_th_ occurred at the simulated altitude of ~2500 m compared to the other two conditions.

## 1. Introduction

Heart rate variability (HRV), i.e., the change in the time intervals between subsequent heartbeats, is an emergent property of interdependent regulatory systems that operate on different time scales to adapt to challenges and achieve optimal performance [1]. Monitoring of the HRV by the time differences of subsequent RR intervals (the time elapsed between two successive R waves of the QRS signal) on the electrocardiogram (ECG) curve has been established as a useful non-invasive tool for the assessment of autonomic nervous system (ANS) activity, particularly parasympathetic (vagal) cardiac regulation [2]. Vagal activity can be mirrored by time domain parameter of the mean of the squares of the successive differences (MSSD) between adjacent RR intervals [3]. It is also associated with the respiratory modulated fluctuation of heart rate (HR) that causes a phenomenon known as respiratory sinus arrhythmia [4].

Physical exercise belongs to significant variables influencing the balance between the sympathetic and vagal activity that predominantly regulates HR with intensity and volume of exercise being the most important factors to consider [5]. Regarding increasing intensity of exercise, the accompanying rise in HR [6] has been attributed to vagal withdrawal at low to moderate exercise intensity, while the further increment in HR is induced by the rise in sympathoadrenal activity [7] as a response to the activation of hypothalamic–pituitary–adrenal axis [8].

HRV indexes related to cardiac vagal activity demonstrate a curvilinear decay in which the indexes decrease with increasing exercise intensity to a certain point at approximately moderate intensity [5]. After this point, the values of the vagal-related indexes can be considered almost zero and the change with a further increase in exercise intensity can be considered as negligible. The above-mentioned point occurring at a particular exercise intensity has been defined as the vagal threshold (T_va_) [9]. One of the potential mechanisms of adverse clinical events may be associated with a rapid loss of vagal activity followed by sympathetic rise during exercise with a higher intensity above T_va_ [10]. It is assumed that the determination of the vagal activity threshold (or HR corresponding with T_va_) would be eventually helpful for subjects who are more prone to health complications entailed with elevated sympathoadrenal activity to minimize the risk of sudden cardiac death [11]. As reduced cardiac vagal activity accompanied with an elevated cardiac sympathetic activation has been associated with an increased risk of malignant ventricular arrhythmias or electrical instability of the heart, individuals with such cardiac complications should avoid intensities of exercise above the T_va_ [12]. On the other hand, the physical exercise up to the T_va_ for with preserved vagal activity is considered to be safe in the light of cardiac stress [9] with habitual physical activity being established to have a cardioprotective effect in cardiac patients [13]. Moreover, T_va_ can be used as an indicator of the optimal exercise intensity suitable for health promotion in normal subjects [14].

Hypoxia, as characterized by a decrease in the inspired oxygen pressure [15], belongs to various environmental stressors which attenuate HRV [16]. The lower pressure of inspired oxygen in inhaled air (either terrestrial or simulated altitude) is associated with the decrease in blood oxygen saturation (SpO_2_) [17]. Decreased SpO_2_ level has been considered to be an important factor that influences the cardiovascular compensation response in hypoxic conditions [18]. As reviewed by Oliveira et al. [19], hypoxia has been capable of generating a decrease in HRV primarily due to attenuation in vagal activity which is accompanied by relative increase in sympathoadrenal regulation. Consequentially, hypoxia leads to acceleration in the HR. This effect appears to be dependent on altitude level and barometric pressure.

Intermittent hypoxic training at simulated altitudes of ~2000–2400 m may improve cardiovascular health and autonomic balance [20]. It should also be stressed that high altitude has become a popular leisure-time destination that is visited not only by healthy individuals but also by increasing numbers of patients with preexisting diseases. Acute hypoxia belongs to relevant factors that may represent a trigger for sudden cardiac death [21]. Pre-exposure assessment (e.g., via simulated altitude) helps to minimize risk and detect contraindications to high-altitude exposure or help with preconditioning [22]. A response to exposition to hypoxic conditions is highly individual; hence, it is also desirable to individualize recommendations related to possibilities of exposition to hypoxia including its duration and FiO_2_ and or degree of personal cardiac health, for example by means of monitoring the changes in the HRV [23].

Taken together, the synergistic effect of the two stress stimuli, exercise and hypoxia, employs assumed influence on the reduction in cardiac vagal activity and a relative rise in sympathoadrenal activity, depending on the exercise intensity and degree of hypoxia Therefore, this study was aimed to determine T_va_ level during increasing physical load in normoxia and two normobaric simulated altitudes (~1500 m and ~2500 m) in a sample of healthy men with a possibility of subsequent suggestions how to define an adequate altitude or exercise intensity for individuals with an elevated risk of cardiac complications. Thus, we were interested in the assessment of responses of selected physiological variables to the combined effect of exercise intensity and hypoxia at the T_va_ level.

## 2. Materials and Methods

### 2.1. Subjects

The sample consisted of 28 healthy male volunteers aged 20 to 30 years. They were recruited among students of sports sciences, and the participation was limited only to non-smokers. The original number of participants (*n* = 30) was reduced by two persons who did not complete all requirements of the study. The somatic and physiological characteristics of the experimental group are presented in Table 1.

This study project was approved by the Ethics Committee of the Faculty of Physical Culture, Palacky University in Olomouc (Czech Republic) under the identification number 80/2018. All subjects gave written informed consent in accordance with the Declaration of Helsinki and filled in a form of “Lausanne Recommendations: sudden cardiovascular death in sport” [24]. The participants were informed in detail about the aims and scope of the study with the possibility to withdraw from the study at any time.

### 2.2. Experimental Protocol

All processes associated with the experiment were performed in the laboratory of exercise physiology at the Faculty of Physical Culture, Palacky University in Olomouc. All measurements were conducted between 8:00 a.m. and 1:00 p.m. Ambient temperature in the laboratory ranged between 22 and 24 °C. The altitude of the laboratory was 260 m above sea level (FiO_2_ = 20.9%). The participants were instructed that factors possibly influencing their ANS should be avoided from two hours (e.g., caffeine or tea intake, eating, specific medications or substances) up to one day (e.g., strenuous physical activities) prior to particular measurements.

The study consisted of five main stages, as described graphically in Figure 1 (see below a detailed description of instruments and techniques used). A delay of at least two days between each measurement stage was respected.

The first stage comprised of anthropometry and body composition measures, measuring of resting HR (HRrest) and HRV, spirometry (vital capacity and forced expiratory volume in 1 s), and the participants underwent maximal incremental test on a bicycle ergometer to assess maximal oxygen uptake (VO_2_max), maximal heart rate (HRmax), maximal power output (Pmax), and ventilation parameters (minute ventilation—VE, tidal volume—Vt, and breath frequency—Bf). All measurements were performed in normoxia.

After the determination of HRrest and HRmax, maximal heart rate reserve (MHRR) was computed for each subject according to a formula MHRR = HRmax − HRrest [25]. Consequentially, we calculated individual heart rates corresponding with 20%, 30%, 40%, 50%, 60%, and 70% of MHRR using a formula HRtarget = HRrest + MHRR × K/100, where K is a percentage coefficient.

Stage 2 was performed to set and record individual power outputs and cadences (pedaling frequencies) needed to reach particular target HRs matching previously computed values of 20% to 70% of MHRR. For this purpose, power output and cadence were continuously adjusted during pedaling to reach required HR firstly at 20% of MHRR and to keep it steady (i.e., ± 2 beats·min^−1^) for at least one minute before continuing to the next HR accommodating 30% of MHRR. This process was performed repeatedly up to HR matching 70% of MHRR. The setting procedure was performed in normoxia. Recorded values of power output (watts) and pedaling frequencies associated with particular target HR at 20 to 70% of MHRR were used to create an individual template protocol for testing in the subsequent three experimental stages performed in normoxia and two simulated altitudes of ~1500 m and ~2500 m.

Stages 3 to 5 followed the same procedure consisting of experimental test on the bicycle ergometer, starting with the preset power output and cadence corresponding with 20% of MHRR determined during the setting process in stage 2. Each step of the experimental test (successively at 20%, 30%, 40%, 50%, 60%, and 70% of MHRR) lasted 3 min (expected sufficient time to reach a steady state within each step [26]), so the total duration of the test was 18 min. Order of the stages 3, 4, and 5 was randomized for each participant using a computer-generated permutation table. This process was not disclosed to the participants as they were not informed about the degree of normoxia/hypoxia in which they performed the testing.

### 2.3. Anthropometrical Measurement

Subjects underwent the basic anthropometrical measurement. Body height was measured by a stadiometer. Body mass, percentage of body fat and fat-free mass were determined using bio-impedance analysis (Tanita BC-418 MA, Tanita, Tokyo, Japan).

### 2.4. Resting Spirometry

The spirometry test with calibrated instruments (Spirostik with Blue Cherry software; Geratherm Respiratory, Bad Kissingen, Germany) was performed to assess individual vital capacity (VC) and forced expiratory volume in one second (FEV_1_). These values were also recounted to the values predicted according to the body surface area (%).

### 2.5. Maximal Incremental Testing

The maximal incremental test was performed on the bicycle ergometer Ergoline 800 (Ergoline, Bitz, Germany). The exercise protocol consisted of an 8-min warm-up period (4 min at 120 W and 4 min at 160 W, both with 70 rpm cadence) followed by 1-min incremental steps, starting at 220 W and increasing by 20 W every minute until exhaustion. Cadence increased arbitrarily following the individualized needs of participants to reach the maximum.

Breath-by-breath ventilation and gas exchange were measured (Ergostik with Blue Cherry software; Geratherm Respiratory, Bad Kissingen, Germany) during the exercise with the data averaged to 30 s for analysis. Gas and flow analyzers were recalibrated before each testing using gases of known concentration and a 3-L calibration syringe.

The following criteria were used to document that VO_2_max was achieved: (1) a lack of increase in VO_2_ upon an increase in work rate, and (2) respiratory exchange ratio > 1.10. VO_2_max was recorded as the highest VO_2_ value in the final 30 s of the test. Heart rate response was measured continuously using a chest strap (Polar Electro Oy, Kempele, Finland). HRmax was defined as the highest HR recorded during the test. The relative value of maximal power output (Pmax) was determined as the highest wattage reached and maintained during the last 30 s of the test, divided by individual body mass.

### 2.6. Hypoxic Chamber

Normobaric hypoxia conditions were created using a hypoxic chamber and HR-1470 generator of hypoxic air (Hypoxie group, Prague, Czech Republic). In this study, two levels of normobaric hypoxia with FiO_2_ = 17.3% and 15.3% were used, which correspond to simulated altitudes of ~1500 m and ~2500 m, respectively. Volume of the chamber was 45.5 m^3^ (length: 7.0 m; width: 2.5 m; height: 2.6 m). The generator separates compressed air into nitrogen and oxygen fractions using a system of a hollow fiber membrane. As an output of this separation process, oxygen-reduced air flows into the chamber. Required FiO_2_ inside the chamber was continuously maintained by a controlling system of inlet/outlet valves and calibrated sensors. Carbon dioxide (CO_2_) concentration was kept under 1500 ppm (0.15%) by its regular airing out after each experimental test. Relative humidity of approximately 30 to 40% was maintained in the chamber by a common commercial humidifier.

### 2.7. Heart Rate and Heart Rate Variability Measurement

RR intervals were measured continuously during the experimental test on bicycle ergometer using a Polar V800 heart rate monitor (Polar, Kempele, Finland) which was proven to be a valid tool for measuring the RR interval [27]. RR records were transferred to a computer using the Polar Flow cloud service. Artefacts (ectopic beats, missing beats, etc.) were identified by visual inspection of RR intervals and simply deleted because the deletion method provided the best overall performance [28]. The MSSD value and average HR value were calculated from a 1-min segment that started 1.5 min from the beginning of the 3-min step (Figure 1).

### 2.8. Oxygen Saturation Measurement

Arterial oxygen saturation (SpO_2_) was measured continuously during the experimental test using a Nonin Avant 4000 pulse oximeter (Nonin Medical, Minneapolis, MN, USA) with a sensor placed on the left middle finger. SpO_2_ was measured at a sampling frequency of 1.0 Hz, and the average value was calculated using a 1-min segment that started 1.5 min from the beginning of the 3-min step (Figure 1).

### 2.9. Ventilation Measurement

Breath-by-breath ventilation was continuously measured (Ergostik with Blue Cherry software; Geratherm Respiratory, Bad Kissingen, Germany) also during all three experimental stages 3 to 5. For further statistical processing, the average values of Vt, Bf, and VE were calculated using a 1-min segment that started 1.5 min from the beginning of the 3-min step (Figure 1).

### 2.10. Rating of Perceived Exertion

Rating of perceived exertion (RPE) was applied as a psychophysiological measure during the three experimental testing stages (10 s before the end of each 3-min step). For this purpose, the revised Borg’s category-ratio scale (0 to 10 scale) was used [29]. The subjects were instructed to express a numerical value for their RPE with the help of text descriptors.

### 2.11. Vagal Threshold Determination

Vagal threshold values were calculated using a custom application written in MATLAB 8.4 (MathWorks, Natick, MA, USA). The algorithm used was described in detail by Botek et al. [30]. The input to the algorithm was a set of 6 MSSD values measured at a six different power output values. In order to be able to process subjects with different levels of physical fitness, power outputs in watts were set individually to correspond to 20%, 30%, 40%, 50%, 60%, and 70% MHRR in normoxia (stage 2). The individual power outputs were kept the same during normoxia/hypoxia stages 3 to 5 and were labelled as the percentage of MHRR in normoxia. The algorithm split the set into two subsets and calculated T_va_ as the intersection point of two regression lines. In the beginning, the first subset contained 2 MSSD values obtained on power outputs corresponding to 20% and 30% MHRR, and the second subset contained 5 MSSD values obtained on power outputs corresponding to 30% to 70% MHRR. Regression lines were calculated using least-square method and a condition was tested whether the second regression line has a decreasing slope and lies above the zero level. If the condition was not met, an MSSD value corresponding to the next output power was added to the first subset, and this MSSD value was removed from the second subset. Regression lines were recalculated and a new intersection point was calculated. This iterative procedure was repeated until the condition was met. An illustrative example of calculating the vagal threshold is shown in Figure 2. The algorithm provided power output value (PO_th_) at which the vagal threshold was found and a corresponding MSSD threshold value (MSSD_th_). PO_th_ was expressed in watts and also approximated as a percentage of MHRR related to normoxia. The threshold values of other variables (HR_th_, SpO_2th_, VE_th_, BF_th_, Vt_th_, and RPE_th_) were calculated by linear interpolation using two consecutive steps in the vicinity of the PO_th_.

### 2.12. Statistical Analysis

To verify that RR intervals in the 1-min segment were stationary, the segment was divided into 30 s sub-segments and values between adjacent sub-segments were compared using an analysis of variance (ANOVA) for repeated measures with three fixed factors (altitude, load, and sub-segment order); however, only the significance of the order factor was considered.

Particular vagal thresholds were determined using the algorithm described above and threshold values of studied variables were calculated. Data normality was verified using the Kolmogorov–Smirnov test and *p* < 0.05 was considered statistically significant. The Kolmogorov–Smirnov test rejected the normality of PO_th_, MSSD_th_, VE_th_, and RPE_th_, so the threshold values between normoxia/hypoxia were compared using a set of 3 separate Wilcoxon tests. Bonferroni adjustment was used to control the Type 1 statistical error, therefore *p* < 0.05/3 was considered statistically significant. Potential predictors of PO_th_ were searched using Pearson’s correlation coefficient.

Data are presented as arithmetic mean ± standard deviation. Statistical analyses were performed using MATLAB 8.4 (MathWorks, Natick, MA, USA) and the STATISTICA 13.4 (StatSoft, Tulsa, OK, USA). A priori power analysis considering a Wilcoxon test was performed using G*Power 3.1.9.7 (Heinrich-Heine-Universität, Düsseldorf, Germany) with parameters *d*_z_ = 0.8, α = 0.017, and β = 0.20. The minimal sample size resulted in 21 subjects.

## 3. Results

The 1-min segments were divided into 30 s segments and ANOVA did not reveal any statistically significant difference between adjacent 30 s segments for HR (*p* = 0.999) and MSSD after the logarithmic transformation (*p* > 0.999). This indicates that a pause of 1.5 min after the start of each step was sufficient to achieve RR intervals to be stationary.

The threshold values of physiological and psychological variables in normoxia and normobaric hypoxia of FiO_2_ = 17.3% and 15.3% are given in Table 2. The hypoxia of 17.3% compared to normoxia did not induce statistically significant changes in all studied variables (all *p* ≤ 0.109) except SpO_2_ (*p* < 0.001). The hypoxia of 15.3% compared to normoxia significantly decreased PO_th_ expressed in watts (*p* = 0.007), PO_th_ expressed as %MHRR in normoxia (*p* = 0.001), and SpO_2th_ (*p* < 0.001). These decreases were also statistically significant when comparing hypoxia of 15.3% with hypoxia of 17.3% (all *p* ≤ 0.001). The threshold values MSSD_th_, HR_th_, VE_th_, BF_th_, Vt_th_, RPE_th_ did not change significantly when hypoxia of 15.3% was compared to normoxia (all *p* ≤ 0.126) or hypoxia of 17.3% (all *p* ≤ 0.066).

Individual differences in PO_th_ between hypoxia of 15.3% and normoxia were significantly (*p* = 0.010) associated with individual differences in VE between hypoxia of 15.3% and normoxia observed at exercise step corresponding to 40% MHRR in normoxia (Figure 3). The differences in PO_th_ were not significantly associated with body mass, body fat, VC, FEV1, Pmax, and VO_2_max (all *p* ≥ 0.059) or differences in SpO_2_, BF, Vt, and RPE (all *p* ≤ 0.234).

## 4. Discussion

The primary purpose of this study was to compare changes in vagal activity during exercise with increasing intensity in normoxia and two simulated conditions of normobaric hypoxia. Previous studies were oriented mostly on assessing the T_va_ level in normoxia [9,14,30] or vagal activity in various hypoxic conditions at rest (see review of Oliveira et al. [19]) so the main research question was focused on possible shifts of selected physiological variables determined at T_va_ level during incremental stepwise exercise.

The results of this study show that T_va_ level did not differ excessively between normoxic conditions (PO_th_ of 114 W corresponding with 45.0% of MHRR in normoxia) and normobaric hypoxia with FiO_2_ = 17.3% (PO_th_ of 110 W corresponding with 43.8% of MHRR in normoxia). Significant changes in T_va_ level occurred at normobaric hypoxia with FiO_2_ = 15.3% (T_va_ at 96 W corresponding with 39.9% of MHRR in normoxia) compared to normoxia and normobaric hypoxia with FiO_2_ = 17.3%. To be more specific, HR at the T_va_ remained comparable (116–118 beats·min^−1^) in all the three conditions while PO_th_ matching the “algorithm-determined” HR_th_ decreased with decreasing FiO_2._ Initial exercise intensity corresponding with the vagal withdrawal in normoxia is in full accordance with previous findings of Botek et al. [30] except for HR at T_va_, which was lower in their study of *n* = 10 age-matching subgroup of men (112 beats·min^−1^). However, they used a treadmill instead of a bicycle ergometer in their experimental protocol. Similarly, Hautala et al. [31] considered approximately 40% of VO_2_max as the exercise intensity, where increased sympathetic activation starts to dominate after vagal withdrawal. Shibata et al. [9] assessed T_va_ in a sample of 43 middle-aged obese women. In their study, the sympathovagal conversion occurred at 114 ± 9 beats·min^−1^. Oshima et al. [14] found in their study of *n* = 63 normal subjects that the HR at the T_va_ was 112 ± 13 beats·min^−1^ when using a ramp exercise test with a bicycle ergometer but their sample was markedly heterogeneous (men and women of various ages). The authors defined the criterion of the T_va_ as MSSD < 25 ms^2^. The values of MSSD in our study varied between 25–30 ms^2^ while no significant differences were found between normoxic and hypoxic conditions.

Vagal activity with exercise intensities above the T_va_ can be suggested as negligible with minimal regulatory impact on the cardiovascular system. Iwasaki et al. [32] performed a study focused on a similar degree of normobaric hypoxia (stepwise decreases from 21% to 15% of FiO_2_) at rest. They found that acute exposure to normobaric mild hypoxia (FiO_2_ ≥ 15%) induced a significant shift of the autonomic balance towards cardiac sympathetic dominance resulting in an increased HR. In general, vagal withdrawal and a relative dominance of sympathetic activity in hypoxia seems to be governed by the arterial chemoreflex [33]. However, there are probably multiple pathways controlling cardiovascular response to hypoxia involved. They include peripheral chemoreceptors, arterial baroreceptors, central nervous system hypoxic response, and lung stretch receptors [34].

Regarding SpO_2_ as a relevant factor influencing the HRV [16], its values at T_va_ significantly differed in our study between all the three experimental stages (normoxia and both hypoxic conditions). According to Rojas-Camayo et al. [35], SpO_2_ in healthy individuals is reduced more evidently at altitudes over 2500 m but the effect of exercise was not taken into account. Woorons et al. [36] suggest that moderate exercise in hypoxia induces a greater arterial desaturation in trained than untrained men with the difference starting at FiO_2_ = 0.154 and below.

One of the acute responses of the body to decreasing SpO_2_ [37] as well as to exercise [6] is an increase in VE which appears to ensure adequate delivery of oxygen to tissues. However, we did not find any considerable differences in ventilation (neither in its components, i.e., tidal volume and respiratory rate) at the level of T_va_. It allows us to claim that exercise intensity about 40–45% of MHRR adjusted for normoxia did not elicit any substantial ventilation changes in hypoxia up to ~2500 m. As shown in Figure 3, lower VE at exercise intensity corresponding to 40% of normoxic MHRR is moderately associated with higher power output. Thus, higher VE would probably mean a decrease in vagal activity and termination of respiratory sinus arrhythmia [38].

As mentioned before, T_va_ can be used as an indicator of the “safe” exercise intensity suitable for health promotion in normal subjects [14]. Shibata et al. [9] claim that T_va_ may also be recommended for obese people who might possess a lower cardiac sympathovagal balance. Shiraishi et al. [39] found that T_va_ was strongly correlated with the ventilatory threshold in a subset of patients with myocardial infarction as well as healthy subjects when using real-time HRV analysis during ramp exercise protocol. Thus, we suggest that our study will be helpful to create a standardized procedure, which would enable setting individualized recommendations for the exercise intensity (or physical activity intensity, in general) for various groups of people in need of HR control during physical load in hypoxic conditions in order to maintain cardiac vagal protection.

As a main limitation of the study, we consider the fact that only hypoxic chamber without the possibility of creating hypobaric conditions was used. Moreover, our participants were exposed only to acute short-term hypoxic conditions so the response might be different if longer pre-acclimatization period was involved. We were also limited in possibilities to assess oxygen consumption during the experimental stages because used system of a gas analyzer and associated software was limited to work accurately in the hypoxic air. A further limitation is that we were oriented predominantly on vagal activity with the omission of sympathoadrenal system activity.

## 5. Conclusions

Exercise intensity corresponding with the withdrawal in vagal activity was ~45% of MHRR in normoxia. Considering the convenient altitude for exercise in hypoxia, PO_th_ did not differ significantly between normoxic conditions and FiO_2_ = 17.3% (simulation of ~1500 m). More substantial decline of the PO_th_ occurred in FiO_2_ = 15.3% (simulation of ~2500 m) when compared to the other two conditions. Consequentially, T_va_ adjusted for normobaric hypoxia with FiO_2_ = 15.3% declined to ~40% of MHRR. Further studies performed in samples with different characteristics according to age, sex, or type of health issues are recommended.

## Figures and Tables

**Figure 1 ijerph-17-07579-f001:**
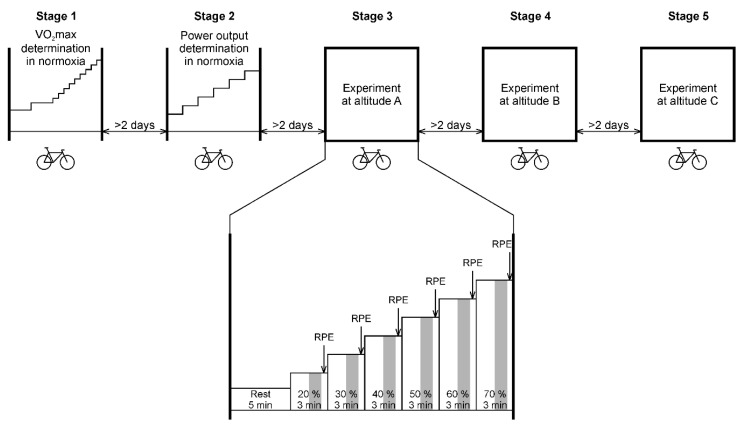
Course of experimental protocol. Altitudes marked A, B, C were randomly assigned to normoxia and two levels of normobaric hypoxia. Gray rectangles denote 1-min segments for calculating average values of heart rate variability, oxygen saturation, and ventilatory measures. RPE: rating of perceived exertion, SpO_2_: arterial oxygen saturation.

**Figure 2 ijerph-17-07579-f002:**
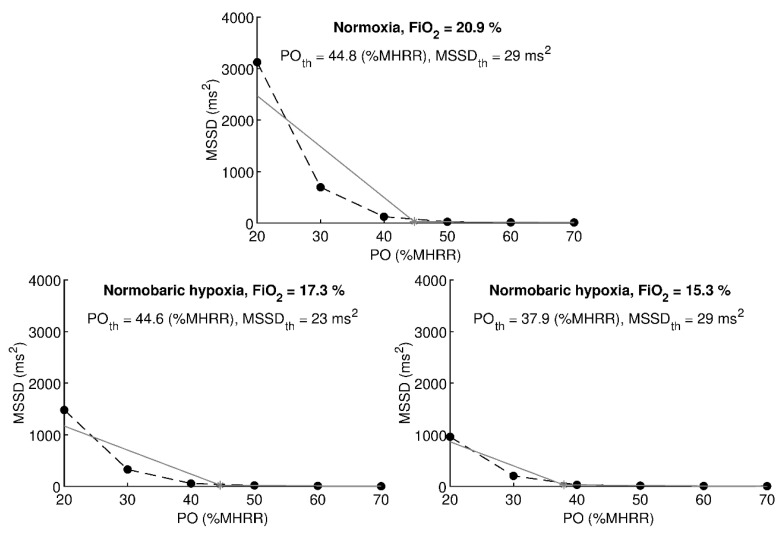
Illustrative example of calculating vagal threshold of one chosen participant at normoxia and two levels of normobaric hypoxia. Black circles and dashed line denote measured data. Grey lines denote regression lines. Grey asterisk indicates vagal threshold point. PO: power output corresponding to a percentage of maximal heart rate reserve in normoxia, MSSD: the mean of the squares of the successive differences between adjacent RR intervals, th: threshold.

**Figure 3 ijerph-17-07579-f003:**
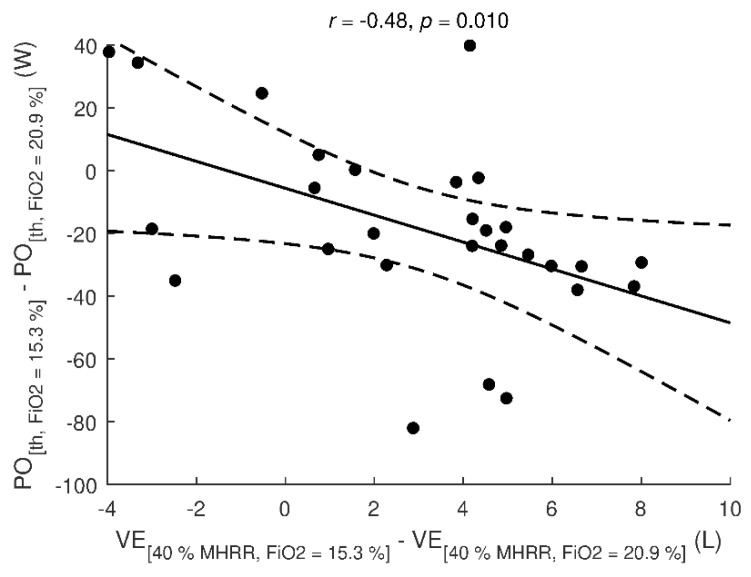
Association between changes in ventilation at step corresponding to 40% MHRR and changes in power output at vagal threshold. Changes were calculated as the value at hypoxia (FiO_2_ = 15.3%) minus the value at normoxia (FiO_2_ = 20.9%). Dashed lines denote 0.95-confidence interval. *r*: Pearson correlation coefficient, *p*: significance of correlation, VE: minute ventilation, PO: power output, MHRR: maximal heart rate reserve, th: vagal threshold.

**Table 1 ijerph-17-07579-t001:** Anthropological and physiological characteristics of the participants (*n* = 28).

Variable	M ± SD	Variable	M ± SD
Age (years)	23.5 ± 2.0	FEV_1_ (L)	4.7 ± 0.7
Body mass (kg)	74.2 ± 6.9	FEV_1_ predicted (%)	108 ± 12
Body height (cm)	176.7 ± 7.2	VO_2_max (ml·kg^−1^·min^−1^)	48.2 ± 6.4
BMI (kg·m^−2^)	23.8 ± 1.9	HRmax (beats·min^−1^)	191.6 ± 7.2
Body fat (%)	10.9 ± 3.5	HRrest (beats·min^−1^)	56.6 ± 5.2
FFM (kg)	66.0 ± 6.2	MHRR (beats·min^−1^)	134.9 ± 7.6
VC (L)	5.3 ± 0.8	Pmax (W·kg^−1^)	4.6 ± 0.5
VC predicted (%)	98 ± 11		

All values were obtained in normoxic conditions. M: mean, SD: standard deviation, BMI: body mass index, FFM: fat-free mass, VC: vital capacity of lungs, FEV_1_: forced expiratory volume in 1 s, VO_2_max: maximal oxygen uptake, HRmax: maximal heart rate, HRrest: resting heart rate, MHRR: maximal heart rate reserve, Pmax: maximal power output.

**Table 2 ijerph-17-07579-t002:** Threshold values of physiological and psychometric variables at normoxia (FiO_2_ = 20.9%) and normobaric hypoxia (FiO_2_ = 17.3% and 15.3%).

Variable	Threshold Value			Comparison		
	20.9%	17.3%	15.3%	17.3% vs. 20.9%	15.3% vs. 20.9%	15.3% vs. 17.3%
	M ± SD	M ± SD	M ± SD	*p*	*p*	*p*
PO_th_ (W)	114 ± 29	110 ± 27	96 ± 32	0.164	0.007	0.001
PO_th_ (%MHRR)	45.0 ± 5.6	43.8 ± 5.0	39.9 ± 5.5	0.109	0.001	<0.001
MSSD_th_ (ms^2^)	30 ± 20	25 ± 17	29 ± 31	0.194	0.567	0.920
HR_th_ (beats·min^−1^)	116 ± 9	117 ± 8	118 ± 9	0.991	0.126	0.066
SpO_2th_ (%)	95.1 ± 1.7	89.9 ± 2.3	85.6 ± 2.9	<0.001	<0.001	<0.001
VE_th_ (L)	34.8 ± 9.5	34.1 ± 10.2	33.5 ± 10.0	0.399	0.255	0.779
Bf_th_ (breaths·min^−1^)	20.3 ± 5.5	19.4 ± 5.2	19.5 ± 5.8	0.150	0.218	0.339
Vt_th_ (L)	1.77 ± 0.44	1.84 ± 0.60	1.78 ± 0.52	0.374	0.884	0.630
RPE_th_ (points)	3.2 ± 1.2	2.9 ± 0.9	2.9 ± 1.1	0.138	0.227	0.732

M: mean, SD: standard deviation, *p*: significance of Wilcoxon test, PO: power output, th: vagal threshold, %MHRR: percentage of maximal heart rate reserve in normoxia, MSSD: the mean of the squares of the successive differences between adjacent RR intervals, HR: heart rate SpO_2_: arterial oxygen saturation, VE: minute ventilation, Bf: breathing frequency, Vt: tidal volume, RPE: rate of perceived exertion.

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
