# Peer review of "Vagal Threshold Determination during Incremental Stepwise Exercise in Normoxia and Normobaric Hypoxia"

_ijerph, 2020, doi:10.3390/ijerph17207579_

Round 1

Reviewer 1 Report

Introduction:

lines 75-77 are not relevant for the study

include study with same aim

Materials and methods:

2. 2. - 2. 6. need to be shorter (lines: 131-198), also 2.11.

Delete Figure 2. and also from discussion lines 365+366.

Reviewer 2 Report

The manuscript describes a novel study with practical implications. The design is sound and the manuscript is well written and depicts the investigation well. Other than a small few minor grammar revisions (i.e. line 31, the word "to" does not seem to be there), this manuscript is fit for publication.

Reviewer 3 Report

Dear authors,

I read with interest your work. In the current study, the authors attempted to measure vagal response to different altitudes. The article is well written; however, I would highly suggest carefully addressing the following points before considering the work.

  1. Heart variability is a useful a non-invasive technique that predominantly reflects parasympathetic (vagal) cardiac ANS regulation. Please expand on how this association was determined. Is this just a relationship or true measurement of vagal regulation?
  2. Can you please clarify what do you mean by this “ Vagal activity can be mirrored by time-domain parameter of rMSSD (square root of the mean of the squares of the successive differences between adjacent RR intervals in ms),  
  3. Line 70-71: Can you please further explain, how mechanistically this happen “hypoxia has been capable of generating a decrease in HRV either by reduction or maintenance of vagal modulation, by sympathetic predominance or even the combination of these responses.
  4. Please clearly define the Vagal threshold (Tw), the definition provided is totally unclear.
  5. It is unclear from the manuscript whether this a real altitude (i.e. natural) or you have artificially provided simulation to the altitude.
  6. Why you have selected your exercise in the range of 20% to 70% of MHRR.
  7. How many RR intervals, you have considered to measure MSSD per each stage. I am assuming that each stage lasted 3 minutes, when did you start capture your measurements at the beginning till the end or the last 15 seconds in each stage. Please clarify
  8. What do you mean by after 1.5 min pause, does each participant had to pause before starting a single stages, this means if we have 6 stages, we ended up have 6 pauses. Please clarify.
  9. The figures you have presented to show the calculations are acceptable, please provide more details on how you come with cutoff points and the regression lines. It is difficult to understand how you had the dotted line and how did you fit the intersection line.
  10. Line 263-2The way you are presenting your results are totally incorrect, you need to provide the results of one-way ANOVA and whether you have done post-hoc adjustments or not. It looks to me like you have run t-tests.
  11. Line 294-295: The results of this study show that Tva did not differ excessively between normoxic conditions (Tva at 45.0% of MHRR) and ~1,500m (Tva at 43.8% of MHRR). Significant changes in terms of vagal withdrawal occurred at a simulated altitude of ~2,500m (Tva at 39.9% of MHRR) when compared to 296 normoxia and ~1,500m. This may be true regarding %MHRR, but MSSD did not show any difference among the three conditions. The main focus of the work was to measure MSSD (ms) and not MHRR.
  12. The biggest point, you need to discuss is that why despite the 3 different altitude your MSSD values did not change among the 3 conditions.
  13. The discussion section is not focused and so long and highly encourage the authors to cut short many of the unrelated ideas. I would highly suggest discussing limitations and future implications of the current work. I am not really sure if you need to highlight other studies that dealt with physically inactive

Reviewer 4 Report

After reviewing the manuscript entitled “Vagal threshold determination in normoxia and normobaric hypoxia” my opinion is that it can be reconsidered after major revision.

Comments:

In general, this is an interesting paper. The authors evaluate the effect of different simulated altitudes (normobaric hypoxia) on HRV-related vagal threshold.during exercise. However, there are some aspects that the authors should take into consideration:

Title.

- The title should be rewritten to reflect that the vagal threshold was evaluated during exercise.

Abstract.

- The acronym “MSSD” must be defined.

Materials and methods.

- The authors must indicate the power of sample size and then discuss about it.

- Clarification is needed regarding the use (or not) of Karvonen’s formula* to calculate the individual HR corresponding to 20-70% of MHRR. As it is indicated in the manuscript (lines 136-139), MHRR was calculated subtracting HRrest from HRmax. Thereafter, the authors calculated individual heart rates corresponding with 20%, 30%, 40%, 50%, 60%, and 70% of MHRR. Nevertheless, assuming the use of these calculations and taking for instance values of HRmax=200 bpm and HRrest= 50 bpm, the target HR corresponding to 20% of MHRR would be 30 bpm, which is lower than HRrest.

*target HR = HRrest + (HRmax – HRrest) x K, where K is a percentage coefficient.

- In my role as a reviewer, I need to check some data regarding stages 1 and 2. Thus, the authors should send (as supplementary files) the following information:

Stage 1: Continuous data-numerical and graphical summaries of the best and worst GXT performances (including variables such as time, watts, rpm, VO2, VCO2, and RQ).

Stage 2: Mean HR, power output (watts) and rpm corresponding to each intensity level (20-70% MHRR).

- Why the authors did not use the Shapiro-Wilk test to check normality (n=28)?

Results.

- Data regarding VO2 (and percentage of the predicted VO2max) during exercise under normoxia and hypoxia conditions should be included in table 2.

Author Response

Please see the attachment. We also have sent the files you have requested as a supplementary material (*.zip) to assistant editor (this editorial system does not allow it) so we hope the supplement will find you well.

Round 2

Reviewer 4 Report

The revised manuscript can now be accepted for publication.